# Tax Fraud Reduction Using Analytics in an East European Country

Tomas Ruzgas [1], Laura Kižauskienė [2], Mantas Lukauskas [1], Egidijus Sinkevičius [1], Melita Frolovaitė [1] and Jurgita Arnastauskaitė [2,*]

[1]  Department of Applied Mathematics, Kaunas University of Technology, 51368 Kaunas, Lithuania
[2]  Department of Computer Sciences, Kaunas University of Technology, 51368 Kaunas, Lithuania
*   Correspondence: jurgita.arnastauskaite@ktu.lt

**Abstract:** Tax authorities face the challenge of effectively identifying companies that avoid paying taxes, which is not unique to European Union countries. Limited resources often constrain tax administrators, who traditionally rely on time-consuming and labour-intensive tax audit tools. As a result of this established practice, governments are losing a lot of tax revenue. The main objective of this study is to increase the efficiency of the detection of tax evasion by applying data mining methods in the East European country Lithuania, which has a rapidly developing economy, by applying data mining methods concerning affluence-related impacts. The study develops various models for segmentation, risk assessment, behavioral templates, and tax crime detection. Results show that the data mining technique can effectively detect tax evasion and extract hidden knowledge that can be used to reduce revenue losses resulting from tax evasion. This study's methods, software, and findings can assist decision-makers, experts, and scientists in developing countries in predicting tax fraud detection.

**Keywords:** tax evasion; fraud detection; data mining; clustering; prediction

## 1. Introduction

Tax avoidance is widely analyzed in scientific research and practical applications. One of the earliest studies on tax avoidance was conducted by Allingham and Sandmo [1], who developed a model of proportional taxation and tax evasion. In this model, the taxpayer assesses the likelihood of an audit and the amount of revenue that can be concealed. Research by Murphey [2] and Schneider [3] shows that tax evasion in Eastern Europe is high, and the shadow economy is one of the largest in Europe. For instance, the shadow economy in Lithuania was 25.8% in 2015, while the European average was 18%. The shadow economy refers to economic activity that is not taxed or regulated, and it often involves tax evasion.

Collecting taxes is essential for providing and maintaining public services, but in practice, many taxpayers may attempt to avoid their obligations, despite recognizing the benefits of paying taxes. This can lead to significant revenue losses for the government. Financial resources must be allocated to identify and address tax evasion, creating a sense of inequality between those who pay their taxes fairly and those who seek to avoid their obligations. One example of such revenue losses is profit shifting and tax revenue losses related to foreign direct investment. To fully understand the scope of the issue, it is important to estimate the scale of these losses and to implement measures to prevent and address them.

Tax audits are a crucial tool for enforcing tax laws in all countries. Research suggests that more frequent audits can help reduce tax avoidance [4]. Random audits are one method

for identifying and expanding the list of potential risks. Practical examples demonstrate that random audits can be valuable for discovering and assessing new risk features. Although this method has advantages, it is also criticized for the uncertainty of its results. The effectiveness of a random audit cannot be predicted in advance, as taxpayers are selected randomly (Federal Republic of German Annual Tax Act Guidelines [5].

Risk management is a continuous process that involves risk identification, analysis, assessment, prioritization, solution searching, and model assessment. Successful audit selection is one of the most critical methods for identifying risks. Audit planning maximizes results while minimizing costs [6]. In the classical approach, auditors give the taxpayer a risk assessment that estimates the potential risk of tax avoidance. In contrast, the modern audit selection process is based on data. Two main data sources are the results of previous audits and taxpayer data, which typically includes economic indicators, such as taxes paid and refunds [5]. Tax administrators generally assess a taxpayer's risk profile and use data-based methods for selecting audits.

Data mining methods are used to prevent potential tax evasion. According to the nature of the available and gathered information, several types of data mining methods are distinguished. Supervised learning methods are applied when having a target variable that statistical methods can predict. Unsupervised learning methods, such as clustering, are used to identify possible anomalies or inappropriate and atypical behaviour.

Although the algorithms of data mining and their various compositions with other innovative analytical methods are widely applied in biomedical, technological, and other sciences [7–10], it is worth noticing that application to economic data was introduced only recently. The first work linking neural networks with economics is the scientific publication of Kuan and White [11], which draws attention to the existing analogues between econometrics and neural networks. Theoretical insights of the denoted research further contributed to the practical work of Maasoumi et al. [12]. These authors demonstrated that the widely quoted macroeconomic lines of Nelson and Plosser [13] can be well modelled using data mining techniques, especially artificial neural networks. The main advantage of these models is that they are better at modelling jumps that significantly deviate from linearity. Considering later applications in the domain of economics, it is important to mention other highly cited research in the field of finance [14], macroeconomics [15–20] monetary politics [21], advertising and marketing [22,23]. Analytical insights on taxpayers' behaviour and more efficient tax collection are topical issues for tax administrators. Methods based on the experience of an expert were used for a long time. Recently, techniques were introduced that allow the automation of processes and easier identification of risky payers. More and more attention is drawn to research works dealing with complex methods [24–27], which give more accurate results.

Scientific articles analyze social, demographic, and financial indicators influencing tax evasion. The selection of different indicators allows for analyzing taxpayers and predicting behaviour based on historical data. The growing amount of gathered data on taxpayers' characteristics and advanced data mining methods allow detailed analysis and identification of risky taxpayers and more accurate tax checks. Therefore, the important question arises of what factors should be chosen and what methods should be used to identify legal entities that avoid taxes.

The novelty of the research is the practical application of data mining methods for the identification of tax evasion of legal entities in one of the European countries: Lithuania. It is a country with rapidly developing economies concerning affluence-related impacts. Lithuania's tax laws and those of other member states align with the European Union's requirements. However, it should be emphasized that the behaviour of taxpayers in Eastern Europe differs from that of other regions of the European Union.

This paper is organized as follows: Section 2 provides a conceptual model for tax obligations' risk management. It shall be based on the guidelines set out by the Organization for Economic Cooperation and Development. Section 3 reviews the world leader software in the field of advanced analytics. Focus on IT market research by Gartner, Inc., and

International Data Corporation. Sections 4 and 5 discuss the data mining techniques applied and the experimental results. The advantages and disadvantages of the methods are discussed by reviewing their use by the tax authorities in this study and the insights revealed by other researchers. The tax administrator's micro-analytics, macro-analytics, and behavioural models create original knowledge that significantly increases the detection of tax evasion. Two examples illustrate this article. Finally, Section 6 provides the main conclusions and future implications.

## 2. Conceptual Model for Tax Obligations' Risk Management

Taxpayer monitoring and control is a strategic risk management tool used by tax administrators in many countries worldwide to ensure taxpayers fulfil their tax obligations. Tax administrators often use limited resources to improve performance management, increase operational efficiency, enhance tax administration, and reduce the prevalence of the shadow economy.

The first task of monitoring and controlling tax activities is identifying taxpayers likely to evade or misapply their tax obligations. The second task is to effectively enforce measures that ensure compliance with tax laws by identifying violations, preventing them, calculating and declaring taxes correctly, and reducing unreported income, tax avoidance, and the size of the shadow economy.

These targeted activities of the tax administrator are part of the risk management model of tax obligation—consolidated activities of the tax administrator (Figure 1) [28].

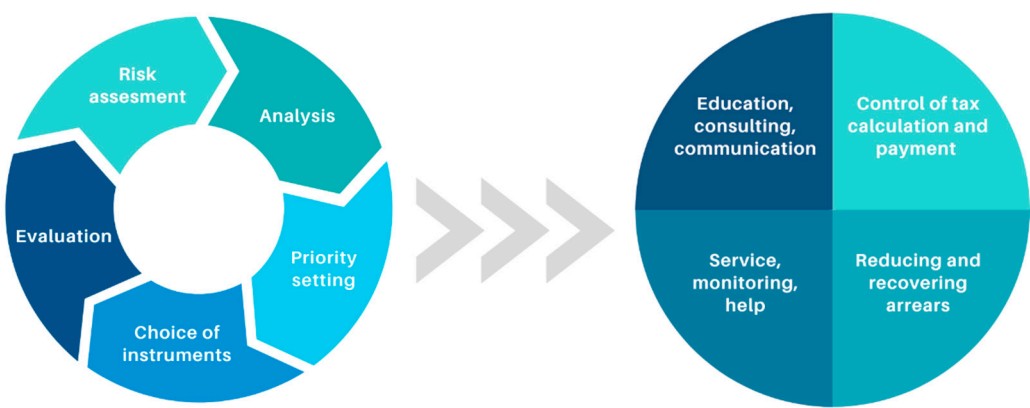

**Figure 1.** Tax administrator's risk management model and objective strategies.

The guidelines (MTC, 2008; MIC, 2009) issued by The Organization for Economic Co-operation and Development (OECD) provide a conceptual model for the monitoring and control of tax compliance, which is also applied by the State Tax Inspectorate of the Republic of Lithuania (STI).

The model presented by the OECD includes the tax system as a whole and splits it into separate elements according to the main administered taxes (in Lithuania, it includes value-added tax (VAT), personal income tax (PIT), and income tax), taxpayer groups or segments, as well as taxation risk factors.

Effective selection and application of targeted taxpayer monitoring and control measures are only possible after complex data analysis covering detailed information on taxpayers' segments and inherent risks.

### 2.1. Taxpayer Grouping

One of the stages of tax risk analysis and assessment is 'taxpayers' grouping or segmentation. Grouping aims to distinguish and characterize taxpayers' groups, identify each group's specific behavioural characteristics, and apply the most effective targeted (targeted to a particular tax group) tax enforcement measures. The division of taxpayers into groups according to certain characteristics and behaviours allows us to better

identify and assess the risk of non-fulfilment of tax obligations and the tendency to avoid these obligations.

The grouping of taxpayers in different countries varies according to different characteristics. For example, the usual practice (applied in Lithuania as well) divides taxpayers into groups, distinguishing them into legal entities and individuals who are subsequently categorized according to the performance criteria (i.e., run economic activities or not) by the scope of activity, number of employees, and others.

In Lithuania, taxpayers are usually divided into the following groups with their characteristic attributes: Major Taxpayers, Medium Taxpayers, Small Taxpayers, Individuals that run economic activities, and Individuals that do not.

In practice, according to the OECD Guidelines on Taxation Grouping, other ways of grouping taxpayers by attributes are also applied in SIT, for example, by:

- the type of economic activity;
- payable fees and their amount (e.g., income tax, VAT, excise duties, and others);
- taxable person specifics (e.g., small businesses, social enterprises, companies established in free economic zones, and others, which are subject to different tax rates and/or taxation rules for corporate tax);
- other characteristics or combinations thereof.

The data on taxpayers who pay the largest taxes to SIT accounts, provided by the SIT website [29], shows that SIT mainly groups taxpayers by the amount of taxes and the sectors of activity (industries). It can also be seen that it separately analyzes the amount of tax paid by taxpayers, whose legal form is a public institution. The list of TOP 500 legal entities (excluding budgetary institutions and public bodies) paying the most taxes for a certain period can also be found in the denoted resource.

Taxpayers can be grouped according to their common characteristics, the amount of taxes paid, and their behaviour.

According to the OECD Guidelines [30,31], taxpayers are also grouped according to the level of compliance with tax obligations. Figure 2 shows four levels of tax compliance:

1. Taxpayers who honestly perform tax obligations;
2. Taxpayers who strive to honour their tax obligations fairly but not always successfully;
3. Taxpayers are generally inclined not to comply with their tax obligations until they are addressed;
4. Taxpayers who maliciously avoid fulfilling tax obligations.

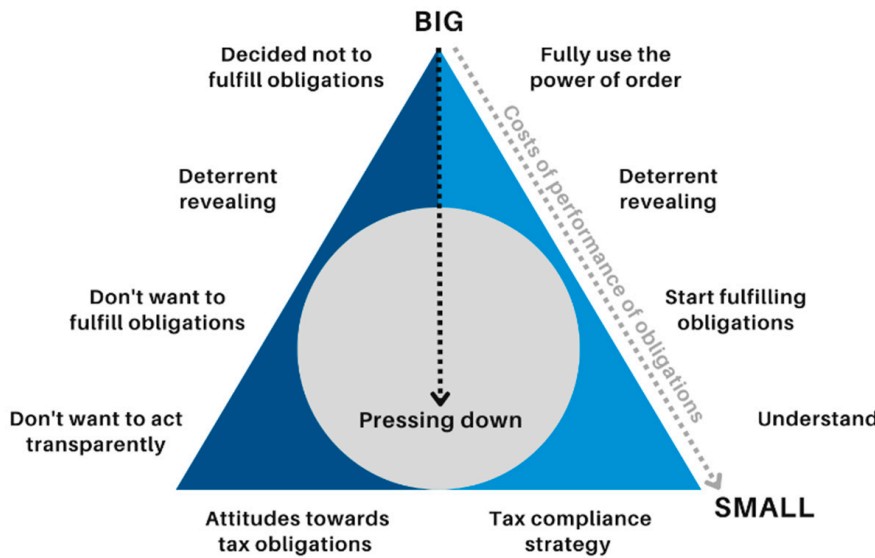

**Figure 2.** Taxpayers' grouping concerning the fulfilment of tax obligations.

The tax administrator analyzes and determines the tendency of each group of taxpayers to fail to fulfil their tax obligations, taking into account the specific type of obligation and the appropriate tax enforcement measures. This assessment applies to each group based on their tendencies to violate the law, make errors, evade taxes, or fail to declare or pay taxes.

Seeking to use taxpayers' grouping for more effective risk management of tax obligation fulfillment, taxpayers are grouped by attributes and behaviour, i.e., taxpayer groups are formed according to several criteria simultaneously. Such "combined" grouping is also observed in Lithuania. For example, during taxpayer control projects, the taxpayers' grouping is performed according to their scope of activity, specific tax obligations, the risk of obligation non-fulfillment, and the tendency to tax evasion or incorrect declaration.

In order to use taxpayers' grouping for more effective risk MANAGEMENT of tax obligation fulfilment, taxpayers are grouped based on their attributes and behaviour, formed according to several criteria simultaneously. This 'combined' grouping is also observed in Lithuania. For example, during taxpayer control projects, taxpayers are grouped according to their scope of activity, specific tax obligations, the risk of obligation non-fulfilment, or even their tendency to engage in tax evasion or make incorrect declarations factors.

Examples of SIT-implemented projects include "Transparent Production", "Responsible Construction", "Lossy Company", "Show without shadows", "Wheels", "Housing without Tax", and "White Gowns", where attributes perform the segmentation of taxpayers (e.g., the nature of the activity) and their behaviour (predisposition to certain behaviour). During these projects [29], the tax administrator intentionally takes control actions in the identified risk segments, where some common behaviours of taxpayers prevail (inclined to fail or maliciously avoid tax obligations).

### 2.2. Taxpayer Grouping

According to OECD and European Union (EU) practice, four types of tax compliance risks are distinguished:

1. registration risk (the person does not register as a taxpayer in time, does not deregister from the taxpayer register, or registers for other than economic purposes);
2. the risk of incorrect tax declaration;
3. the risk of delayed tax declaration when filing tax returns is delayed, or declarations are not provided at all;
4. tax payment risk, where the payment of taxes is delayed or the fees are not paid.

Based on the risks mentioned above, the Lithuanian tax administrator distinguishes specific risks [28] that have the greatest impact on the collection of income of the Republic of Lithuania. They are identified as urgent threats, leading to the unsuccessful collection of planned state and municipal budgets and fund income, i.e.,:

1. avoidance of registration of ongoing economic activity;
2. avoidance of taxpayer registration and registration for other than economic activities;
3. avoidance of declarations of income and taxes;
4. incorrect application of illegal use of tax relief;
5. avoidance of labour-related taxes;
6. delay in declaring taxes;
7. non-payment of taxes.

### 2.3. Tax Risk Analysis Model

The European Commission's recommendations [32] propose using an analysis model to identify and assess tax risks, which involves three stages of assessment. The process begins with an assessment of overall risk and then moves on to individual taxpayers, culminating in the evaluation of their specific risks (see Figure 3).

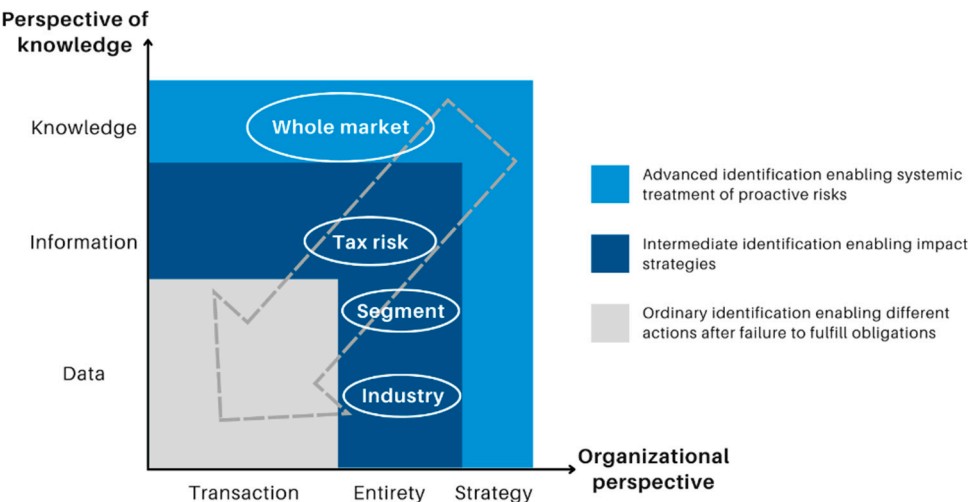

**Figure 3.** Tax risk analysis model (risk identification levels).

*Square 1—Strategic Focus Level*

A review of common risk areas for taxpayers is carried out at this stage. It identifies common tax risks—these may be risks that are already known to the tax administrator (i.e., identified in taxpayer control procedures in the past, the tax administrator is already monitoring a certain level of these risks) or risks that arise because of tax law changes, and introduced or repealed tax reliefs, which the tax administrator can only predict. Identifying common risks considers demographic, economic development, and regional differences. More detailed risk analysis is not carried out at this stage (i.e., it is not assessed which taxpayers are exposed to this risk, in which sector they are exposed, their potential extent or impact on the collection of budget revenue, and others).

*Square 2—Aggregated Focus*

At this level, specific risk areas and groups of risky taxpayers (business sectors) are identified: aggregate–complex analysis of taxpayers' and tax risks is performed using various measures (e.g., analytical software or other tools). The main objects of such analysis are tax risk (risky activity) and risky taxpayer. This analysis provides aggregated information on taxpayers' segments and inherent tax risks.

*Square 3—Case Focus*

At this stage, individual taxpayers' assessment and risks are carried out. They identify and assess a particular taxpayer's risk results from a successful stage 2 analysis. At this stage, the tax administrator is in direct contact with a taxpayer and is subject to appropriate tax risk mitigation measures (information, tax investigation, and others).

This level provides an opportunity to apply analytical, scientific, mathematical, and forecasting methods most effectively and to obtain better results by addressing taxpayer segmentation and risk assessment problems monitored and controlled by the tax administrator.

### 2.4. Evaluation of Key Risks at the Aggregated Analysis Stage

As evidenced by the tax risk analysis model and strategic activities, data analysis is one of the cornerstones of implementing a successful risk analysis model and control actions.

The overview of the current situation outlines that the tax administrator continuously performs data analysis based on taxpayers' monitoring and control. Based on the acquired results, the segmentation of taxpayers, according to their characteristics, behaviour, attitude towards tax obligations, and others, is carried out. The decision upon applying impact measures on the purified segments is taken.

In order to significantly improve the analysis of risk management for non-compliance with tax obligations, it is necessary to enhance and expand the capabilities of the tax

administrator to carry out research using complex scientific, econometric, and statistical methods. This will enable forecasting taxpayer behaviour within specific segments and their potential impact on the state budget. It will also allow for analyzing large data sets on taxpayers, transactions, tax-related documents, and more. Moreover, it will provide the ability to model taxpayers' behaviour and assess the impact of strategies and measures on reducing risks.

### 2.5. Measures to Manage the Risk of Tax Noncompliance

Most effective measures to reduce the risk of tax noncompliance are identified while considering the essential risks and their management measures, different behaviour of taxpayers, and their groups.

The selection of risk reduction measures for taxpayers with different behaviour characteristics is based on the main principle that honest taxpayers are being helped by informing and advising them. At the same time, tax evasion is being controlled and monitored. The scheme below (see in Figure 4) illustrates the application of risk mitigation measures according to the taxpayer's behaviour [28].

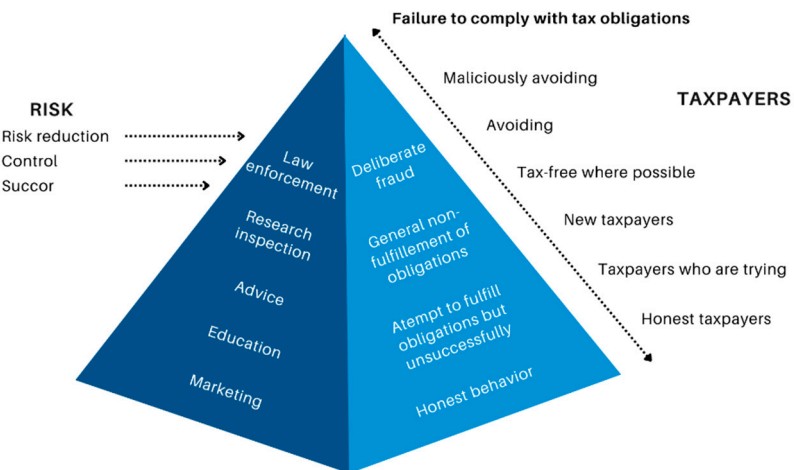

**Figure 4.** The execution of tax obligations [28].

Depending on the problem (risk) and the taxpayer's behaviour, the tax administrator seeks to enforce tax compliance measures that would achieve the highest level of tax compliance at the lowest cost and guarantee the greatest impact on the entire sector, i.e., the optimal control or monitoring measures. Thus, assessing the effectiveness of taxpayers' control and monitoring activities and selecting optimal control or monitoring measures, including determining their scope and duration, is necessary.

The effectiveness of control or monitoring tools can be measured in different ways, for example:

- Taking into account the indicators for assessing the efficiency of taxpayer control processes and evaluating the effectiveness of control or monitoring actions in specific sections (e.g., the execution time of control or monitoring actions, number of irregularities detected, amount of charges charged, amount of fees received, the change in indicators for the sector, where the actions were applied, and others);
- Comparison of actual results of control or monitoring actions with expected results (e.g., comparing expected results with actual results at the end of the measure upon completion of control or monitoring);
- Comparing the procedures for completing specific control actions established by law (for example, a tax investigation ends up with a recommendation for the taxpayer to pay an additional fee and to verify the declarations, while the tax verification is already completed in an act where the additional charge is calculated and recorded as an obligation after subsequent procedures), the duration of actions (established by

legislation, actual average, and others) as well as the results of the application of the different tendency to offend against taxpayer segments;

- Identify and track indicators to assess the effectiveness of taxpayer control and monitoring actions, such as key performance indicators;
- Depending on the possible ways of evaluating the effectiveness of control or monitoring actions, the control or monitoring activities results can be compared with the control group at different periods, and different segments and differences from target values can be assessed. It should be noted that the effectiveness of taxpayers' controls or monitoring activities can only be assessed by providing high-quality, comprehensive, and accessible information about the control and monitoring tools applied and the results acquired. The selection of optimal taxpayer control or monitoring tools and the scope and duration determination is only possible with the above-denoted information and complex data analysis.

When assessing the choice of optimal control or monitoring tools, it is important to consider the most frequent violations by taxpayers, tax risks arising from specific taxpayer groups or sectors, their behavioural modelling, risk assessment, and the effectiveness of control or monitoring measures. Several control actions can potentially be applied if a particular group of taxpayers is characterized by the identified and assessed risks. The optimal measures can be selected based on assessing the effectiveness of control or monitoring tools and defined performance indicators. For example, if a taxpayer segment is identified as having a high risk of fraud in the VAT area, it is subject to operational checks. However, one of the essential features of the identified individual cluster is bankruptcy risk. After assessing the effectiveness of operational checks against bankrupting legal entities, another control action or combination of control actions can be selected (such as tax inspection and appeal to law enforcement authorities).

### 3. Software and Its Functionality

According to Gartner Inc.'s 2021 Magic Quadrant for Analytics and Business Intelligence Platforms, the world's leading information technology research and consulting provider, SAS, remains in the visionary quadrant. This is partly due to SAS's rapidly increasing investment in augmented analytics and its global reach and vertical-specific solutions, as seen in Figure 5. In 2022, SAS received special recognition from Gartner for "market-leading" data visualization with the launch of a new software, SAS Viya 4. However, Gartner questions the limited adoption of the SAS solution and its bundled pricing model (see Figure 6).

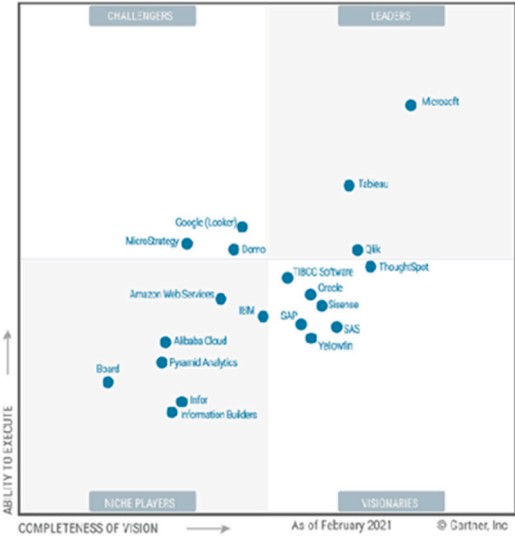

**Figure 5.** 2021 Gartner Magic Quadrant for Analytics and Business Intelligence Platforms.

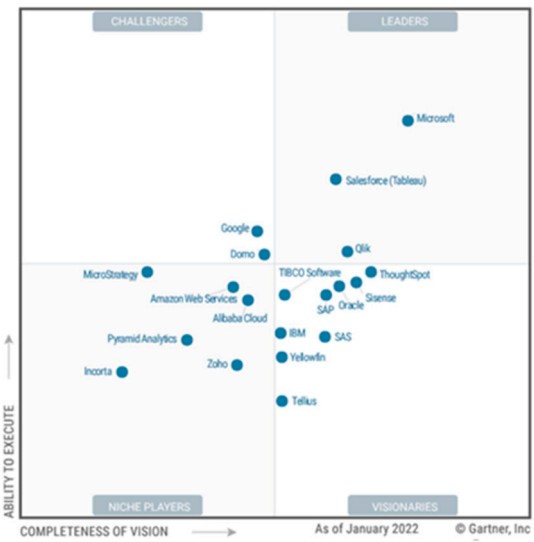

**Figure 6.** 2022 Gartner Magic Quadrant for Analytics and Business Intelligence Platforms.

Other tax administrators also use SAS's analytical, scientific, mathematical, and forecasting software. For example, the Estonian Tax and Customs Board uses SAS to deal with tax evasion problems, such as VAT abuse, tax evasion, vouchers, and others (Veermäe, 2015). SAS allows solving these tasks using creative analytical models and ensures better decisions, more accurate selection of candidates for inspection, and identification of violations of tax obligations.

SAS software is based on multi-tier architecture—the SAS® Intelligence Platform, which consists of the following levels: databases, activity logic, and user (see Figure 7).

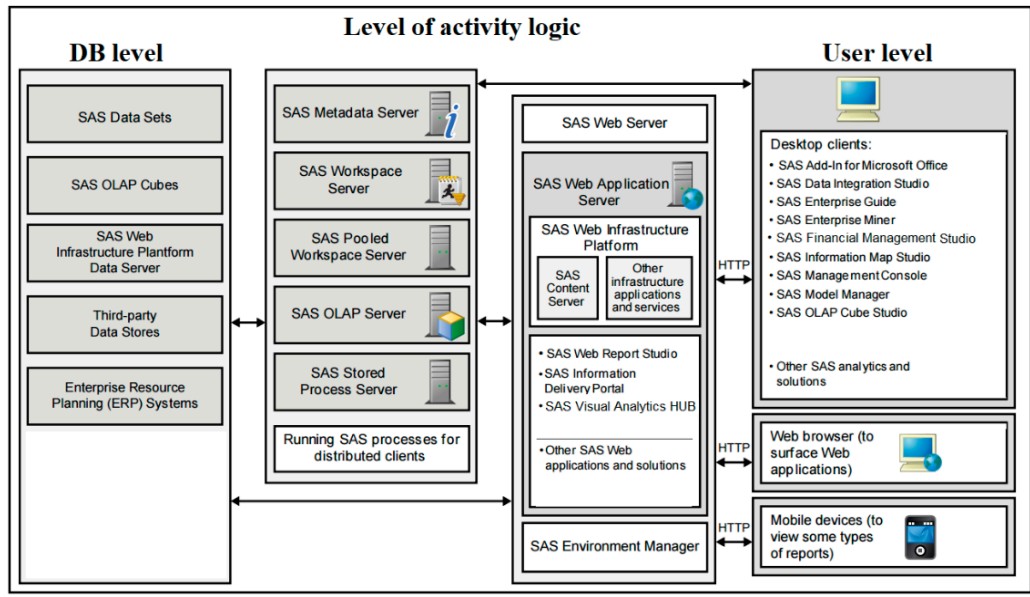

**Figure 7.** SAS Intelligence Platform Multilevel Architecture.

SAS software components, internal and external integration, are implemented in SOA or equivalent technologies. For example, internal and external communication with SAS Metadata Server is based on the XML response principle in the SAS Open Metadata Architecture interface; communication with SAS from external systems can be carried out using the simple object access protocol (SOAP) protocol. Furthermore, SAS can call itself through the SOAP protocol from external sources.

All SAS software is based on well-known technologies and standards in the market, such as SAS Foundation (main module), is created in C language, and SAS clients are created using Java, C#, and VB languages. SAS web application server is based on VMware vFabric tc Server. The following protocols are used for communication: SAS Bridge, SAS OMI, JDBC, HTTP/S, XML, SOAP, JSON, and others. APIs allow SAS to be invoked from external applications running on Java, C++, C#, VisualBasic.Net, and Delphi.

SAS software has its programming language (SAS language). It supports a whole range of technologies and protocols (SAS/ACCESS, OLEDB, OLEDB for OLAP, XML, SOAP, JSON, LDAP/S, message queues, and HTTP/S), which ensures SAS communication with other systems and vice versa; other systems can communicate with SAS (JDBC, OLEDB, OLEDB for OLAP, XML, SOAP, JSON, and HTTP/S). SAS also has special tools for creating data models in physical (SAS® language, SAS® Enterprise Guide®, SAS Data® Integration Studio) and logical levels (SAS® Information Map Studio, SAS® Financial Management Studio). All the denoted technologies provide the ability to create the necessary data models and software solutions independently.

The software is based on an architecture that provides unlimited scalability, i.e., the server where the software operates may expand its capabilities or can be replaced by multiple servers that work together, ensuring that the tasks are parallel without changing the software.

Performance analysis component repository, i.e., the SAS® Metadata Server, consists of three layers:

- Database layer—SAS® Metadata Server Database;
- User layer—SAS® Management Console. SAS® Management Console is the system end-user tool for viewing and modifying the activity logic model using standard activity terminology;
- SAS® Metadata Model for Activity Logic. SAS® Metadata Model is a hierarchical object model defining metadata types and their associations. The SAS® Metadata Model links the database layer to the user layer;
- Software performance analysis component consists of;
- Forecasting module—SAS® Financial Management;
- Free-form analysis module—SAS® Enterprise Guide®;
- View module—SAS® Visual Analytics;
- Standard report generation and distribution module—SAS® Web Report Studio;
- Key performance indicators and their hierarchies module—SAS® Information Delivery Portal and SAS® Web Report Studio;
- Microsoft Office integration module—SAS® Add-In for Microsoft Office.

The software has an advanced analysis component comprising two products—SAS® Enterprise Miner™ and SAS® Visual Analytics. These products have different uses and are intended for different user groups. SAS® Enterprise Miner™ is designed for advanced analysts and users with strong mathematical–analytical skills. SAS® Visual Analytics is designed for data analysis, "friendly" analytics, and reporting. SAS® Visual Analytics is designed for a wider range of users and does not require deep statistical knowledge from users. SAS® Enterprise Miner™ is based on the SEMMA (Sample, Explore, Modify, and Assess) methodology that allows for performing primary and statistical data mining, data preparation, and model accuracy estimation actions.

## 4. Adapting Analytical Methods to Risk Management of Tax Obligations

### 4.1. Data Grouping and Clustering Methods

The concept of grouping in data mining refers to tasks that identify certain patterns in data, usually without prior knowledge of the structures hidden in data. The concept of grouping describes methods where data records are combined according to their similarity. Data records may include, for example, taxpayer's descriptions. In this case, all similar taxpayers would be merged to maximize the difference between various groups acquired

based on this method. Usually, the groups are formed according to the clustering methods described below.

Clustering is a data analysis method for grouping data into previously unknown groups or clusters. Therefore, clustering is an important part of data analysis that helps to reveal data structure. The objective of clustering can also reduce the amount of analyzed data. Grouped data are analyzed separately. The main purpose of cluster analysis is to divide objects so that the difference within clusters is as small as possible, while between clusters is as large as possible. The selection of a particular method and the interpretation of the results depends only on the researcher.

### 4.1.1. Data Grouping

Tax administrators looking for exclusive ways to monitor and control taxpayers must identify and group the supervised individuals. Grouping is a certain research process that involves dividing the objects under investigation into groups interrelated in content or application tasks. In other words, it is partitioning according to their needs, peculiarities, and behavioural characteristics. These groups may require different monitoring and control measures, have different scale and nature issues, and may be subject to various risks. Grouping can be geographic, psychographic, demographic, and behavioural.

The analysis based on this method ensures that the tax administrator devotes more importance to the target taxpayer and the mutual benefit, the needs are defined more precisely, and satisfaction is more efficient. Countries that use analytics for taxpayers' grouping are subject to lower control costs, while tax revenue estimated per full-time equivalent (Dohrmann and Pinshaw, 2009) is higher up to 10 times.

Grouping can be an effective strategy for managing tax risk when performed deliberately and thoughtfully. However, ensuring that the grouping process is based on reliable and comprehensive data is important. To achieve this, it is necessary to conduct a thorough analysis of the taxpayers and their needs and characteristics. The process of grouping typically involves three main steps:

- a study is being carried out to find out the different characteristics and needs of taxpayers;
- taxpayers are combined according to their characteristics and needs (the process is complex because the same environment can be grouped in various ways, often leading to the discovery of new taxpayer groups. This step ensures a more accurate assessment of taxpayers' wishes and provides targeted information);
- the most appropriate (target) groups are selected, and irrelevant or inappropriate groups for the tax administrator's strategic goals are rejected.

Although grouping can be carried out differently, it is necessary to anticipate trends, their alternatives related to resources, the variability of functions performed, and taxpayers' actions.

### 4.1.2. Hierarchical Clustering

Cluster analysis is a set of methods for creating a practical and informative classification of an unclassified data set. It is a statistical method for identifying homogeneous objects (taxpayers, declarations, VAT invoices, and others) or observation groups (clusters). There are two main classes of cluster analysis methods: hierarchical and non-hierarchical.

Hierarchical clustering methods are such clustering methods that determine the overall structure of interdependencies between all clusters and only then identify the optimal cluster number.

Hierarchical methods are further subdivided into agglomerative and divisive methods. By applying agglomerative methods, all observations are initially treated as separate clusters. In the first step, two observations are clustered together, and in every other step, a new observation is connected to an existing cluster, or two clusters are merged. The process is repeated until one cluster remains.

Hierarchical cluster analysis is a method for grouping objects or records "close" to each other.

The $n$ objects to be clustered are denoted by the set $\chi$ where $x_i$ is the $i$th object:

$$\chi = \{x_1, x_2,\ x_3 \ldots x_n\}. \tag{1}$$

A partition, $\Gamma$ of $\chi$, divides $\chi$ into subsets $\{C_1, C_2 \ldots\ C_m\}$ that satisfy the following expressions:

$$C_i \cap C_j = \varnothing, \tag{2}$$

for all $i$ and $j$ from 1 to $m$, $i \neq j$, where $\varnothing$ is the empty set,

$$C_1 \cup C_2 \cup \ldots \cup\ Cm = \chi, \tag{3}$$

the union of all clusters results in the total quantity of all $n$ objects.

Hierarchical clustering is often presented graphically using a diagram called a dendrogram, denoting the relationship between each cluster and the order in which the clusters were combined. Hierarchical clustering methods differ according to their calculation methodology, complexity, and data to be clustered.

Clustering offers an alternative way to quickly find the required information. Clustering methods, unlike search queries, group the objects (taxpayers, declarations, invoices, and others) in common groups according to their characteristics (the number of employees in the company, the head of the company and/or family members of the chief financier) company, family status, and others) despite the presence/absence of search criteria.

Hierarchical clustering methods differ according to several criteria: their calculation methodology, complexity, and the data to be clustered. There are many ways to calculate the distance between two clusters: the average linkage method, the centroid linkage method, the complete linkage method, the density linkage, the single linkage, and Ward's method.

Hierarchical methods are not very practical for clustering large data sets, but they are useful because there is no need to evaluate the number of clusters. The layout order does not affect hierarchical methods as well.

The hierarchical clustering of taxpayers can be based on the taxpayers' characteristics (e.g., individual, business), debt characteristics (value, duration of debt), and the level of risk or the complexity of debt recovery. Applying hierarchical clustering, the work published by Raub and Chen [33] presents the hierarchical clustering example conducted on several parameters: US-based companies that pay taxes both in the US and in other countries, the ratio of US and foreign tax, dividend ratio to total income, and seven other significant features.

Another work by Liu et al. [34] also reveals a case of application in China where a hierarchical grouping method improves the effectiveness of tax verification when a clustering result is compared to a known tax template.

4.1.3. Center-Based Clustering

If the number of observations is high, then the application of hierarchical clustering methods becomes obsolete. In such cases, non-hierarchical methods, such as $k$-means clustering or centre-based clustering, can then be used.

Given the dataset $\chi = \{x_1, x_2,\ x_3 \ldots x_n\}$ with $n$ data points, each point $x_i$ is a $d$-dimension vector $(x_{i1}, x_{i2} \ldots x_{id})$. We form $n \times d$ matrix $x$. A standard $k$-means clustering algorithm includes the following steps:

1. Cluster centroids initialization: randomly choose $k$ different points as initialized centroids $C_1, C_2 \ldots\ C_k$ for $k$ groups, where $C_k$ is $d$-dimension vector $(C_{k1}, C_{k2} \ldots\ C_{kd})$, $k \in [K]$;
2. Repeat the following until the stopping criterion:

   (a) For $i \in [n]$, $k \in [K]$, compute the Euclidean distance between point $x_i$ and centroids $C_k$ by

$$X_{ik} = \sqrt{\sum_{j=1}^{d}\left(x_{ij} - C_{kj}\right)^2}. \tag{4}$$

(b) Assign each data point $x_i$ to the closest cluster $m_i$ for $i \in [n]$. This can be carried out by computing $k_i \leftarrow \mathrm{arg}min\{X_{i1}, X_{i2}, \cdots, X_{ik}\}$ firstly, and then generate a $k$-dimension one-hot vector $b_i$ where '1' indicates the $k_i$-th component of vector $(X_{i1}, X_{i2}, \cdots, X_{ik})$. We form $K \times n$ matrix $B$ such that the $i$-th column of $B$ is the one-hot vector $b_i$. Let $m_k$ be the $k$-th row of $B$.

(c) Recalculate the average of the points in each cluster. For each cluster $k \in [K]$, compute new cluster center with

$$\varphi_k = \frac{m_k \cdot x}{\chi_k},\tag{5}$$

where $\chi_k = \sum_{i=1}^{n} m_{ki}$ is the point number of $k$-th cluster.

(d) Check the stopping criterion and update the new cluster center with the average. For each $k \in [K]$, compute the Euclidean distance between $\varphi_k$ and $C_k$ at first, and then the squared error can be computed by

$$e = \sum_{k=1}^{K} e_k = \sum_{k=1}^{K} \sqrt{\sum_{j=1}^{d} \left(\varphi_{kj} - C_{kj}\right)^2}.\tag{6}$$

Given a small error $\varepsilon$, if $e \geq \varepsilon$, then update $C_k$ with $\varphi_k$. Otherwise, stop the criterion and output $\varphi_k$.

### 4.1.4. Spherical K-Means Clustering

Many clustering methods work well on spherical clusters. If clusters are in a compressed ellipsoid and close to each other, these clusters can be transformed from elliptical to spherical-like forms. During the transformation, canonical discriminatory analysis is performed on the data. K-means clustering performed on the modified data in the new coordinate system is called spherical k-means clustering.

As noted, if we know the within-cluster covariance matrix, non-spherical clusters can be transformed to make them more spherical. Normally the within-cluster covariance matrix $W$ is calculated using the equation

$$w_{jk} = \frac{1}{n-q} \sum_{c=1}^{q} \sum_{i=1}^{n} d''_{ic} \left(x_{ij} - \overline{x}_{cj}\right) \left(x_{ik} - \overline{x}_{ck}\right),\tag{7}$$

where the classifier, $d''_{ic}$ is
$d''_{ic} = 1$ if observation $i$ is in cluster $c$;
$d''_{ic} = 0$ otherwise.

However, $W$ can more conveniently (without requiring a mean) be calculated as

$$w_{jk} = \frac{1}{n-q} \sum_{i=2}^{n} \sum_{h=1}^{i-1} d'_{ih} \left(x_{ij} - \overline{x}_{hj}\right) \left(x_{ik} - \overline{x}_{hk}\right).\tag{8}$$

In this case, the classifier, $d'_{ih}$ is defined as

- $d'_{ih} = \frac{1}{n_c}$ if observations $i$ and $h$ are in cluster $c$;
- $d'_{ih} = 0$ otherwise.

One of the advantages of this method is that there is no need to calculate distances between all pairs of subjects. Furthermore, this method is more efficient and practical when dealing large amounts of data.

Taxpayer clustering through centre-based methods can be based on the use of 'taxpayers' characteristics, their debt, the level of risk, or the complexity of debt recovery. K-means clustering was applied to segment Chilean VAT payers according to their char-

acteristics [35]. Various examples of the application of k-means clustering for solving tax evasion and anomaly detection problems are provided by other authors as well [36].

*4.2. Classification Methods*

Classification is the assignment of objects to predefined classes. Classification is based on regularities specific to the characteristics of individual groups. Objects are assigned to classes using classification rules. Classification rules can be given but are usually formed based on data. The data used to create the classification rule is called a training sample. Therefore, classification is a supervised learning technique. Classification refers to predicting whether a given object belongs to a particular class.

4.2.1. Decision Trees

In practice, there are many options to consider when making decisions. It is important to define each option and investigate its use's consequences. It is usually impossible to determine precisely all the conditions that influence the outcome of the decision. Occasionally, it is possible to find the probabilities that characterize the combination of some conditions or, at least, the law of distribution of the random size associated with the total number of those conditions. However, there are several situations where the solution has to be chosen, and probability theory cannot be applied. In such situations, there is a need to look at a set of possible solution options, gather as much information as possible on each of them, calculate (if possible) the consequences of each decision, and propose a methodology for selecting one of the possible options, or at least provide a logical analysis of the set of decisions. All the solutions associated with each other in different ways are depicted as branches enabling the identification of possible results. Such a view is called a decision tree diagram.

The basic idea of the algorithm is derived based on the given weak learning algorithm and training set

$$(x_1, y_1), (x_2, y_2), \ldots, (x_m, y_m). \tag{9}$$

First, initialize the distribution of the training set $D_1(i) = \frac{1}{m}$, then perform *T-round* training. In the *t*-th cycle, the weak learning algorithm is trained under the weight $D_t$ to obtain the weak classifier $h_t$.

At the same time, calculate the error rate of the weak classifier under the weight $D_t$:

$$\varepsilon_t = \sum_{i=1}^{N} D_t(x_i)[h_t(x_i) \neq y_i]. \tag{10}$$

Weight is updated with the error rate:

$$D_{t+1}(i) = \frac{D_t(i) \exp(-a_t y_i h_t(x_i))}{Z_t}. \tag{11}$$

When the $\alpha_t = \frac{1}{2} \log\left(\frac{1-\varepsilon_t}{\varepsilon_t}\right)$ is satisfied; $\varepsilon_t$ is the error rate of a weak classifier $h_t$ under weight $D_t$, and the classifier is satisfied when $h_t(x_i) = y_i$, $y_i h_t(x_i) = 1$; otherwise, $y_i h_t(x_i) = -1$, and $Z_t$ is the normalization factor. The final output strong classifier is

$$H(x) = sign\left(\sum_{i=1}^{T} a_t h_t(x_i)\right). \tag{12}$$

*Random forests*. A classification tree is a tool that uses a tree-type chart to assign the given data to the appropriate tree branch. The data are fed into the tree's root node, and by applying filters and thresholds, it is sequentially imposed to the respective branch until it is finally assigned to one of the classes. The random forest consists of many decision trees. Here, a data vector is assigned to the tree root nodes (inputs), assigned to the respective

tree branches until one of the classes matches. After classification in each tree, the results are "voted" throughout the forest leading to the data attribution to a particular class.

The decision trees allow:

- Identify objects (e.g., taxpayers) according to their potential dependence on a particular classification group;
- Assign objects (e.g., taxpayers) to a specific category, such as low, medium, and high-risk groups;
- Predict future events, such as tax evasion, based on the model that was developed;
- Compress a large group of available data, leaving independent variables with only a statistically significant impact on the dependent variable prediction;
- Identify interactions between individual test groups.

The decision tree model uses classic statistical criteria. However, the vivid representation of analytical results in the form of classification or decision trees makes it easy to understand variables' hierarchical dependence and identify specific categories.

Additional information makes it possible to use not a pessimistic or optimistic estimate but a more reasonable criterion: the average of the result. However, the probabilities used in the decision algorithm are often subjective or inaccurate, as they are estimated approximately and based on a small number of tests. However, these probabilities can be adjusted. Such a method is based on the application of the Bayesian formula.

There is no doubt that the decision-maker can reduce the uncertainty of the situation by using additional information, and thus, finding a better solution. This additional information is obtained by performing a particular experiment or processing data related to such a situation. The probabilities of events can then be adjusted.

A decision tree makes it easier to analyze the task. Due to this representation, the consequences of each decision become obvious, some data can be meaningfully changed, and the importance of additional information can be evaluated. All calculations for making decisions are rather elementary and therefore available to a specialist analyzing taxpayers' behaviour and assessing their riskiness. Therefore, it is appropriate to make decision trees and use sound conclusions.

T characterizing known taxpayers and attributing their distinct groups of behaviour enables assessing the behaviour of other individuals. Gonzales and Velasquez (2013) published a Chilean tax administrator methodology based on decision trees that define related behavioural rules for detecting fraud and non-fraud cases. Another work by Gepp et al. (2012) presented methods, including decision trees, for solving financial fraud detection and other data mining tasks in the USA.

### 4.2.2. Logistic Regression

All regression models are designed to describe the dependence of one variable on other variables. Some phenomenon (e.g., the tendency to tax evasion, GDP, taxpayers' activity, and others) is being investigated, and their dependencies must be identified. The difference between logistic and linear regression lies in the type of the dependent variable being modelled. In linear regression, the dependent variable is considered continuous, whereas, in logistic regression, it is categorical, i.e., discrete. The logistic function, an accumulated logistic distribution, evaluates probabilities depending on one or more independent variables.

Suppose $n$ independent observations $(x_i, y_i)$ are modeled by a logistic model

$$\log\left(\frac{P(y_i = 1)}{1 - P(y_i = 1)}\right) = x_i'\beta, \ i = 1, 2, \ldots, n. \tag{13}$$

Then $y_i \sim \text{Bernoulli}(\pi_i)$, where $\pi_i = P(y_i = 1) = \frac{\exp(x_i'\beta)}{1 + \exp(x_i'\beta)}$.

The application of logistic regression can be categorized according to the values of the target variable. More specifically, it can be described as:

- Binary logistic regression. The target variable is binary. For example, while investigating what determines the attitude towards tax evasion (need–no need), we want to find out what the decision to fake financial accounts depends on;
- Multiple logistic regression. The target variable is categorical, but it acquires more than two values. We want to find out what influences the choice of the pension insurance fund and what determines the voting priorities (which party to choose);
- Ordinal logistic regression. The target variable is ordinal, with values indicating an increasing (decreasing) amount of some property. For example, to decide what determines the evasion of some tax, all taxpayers are divided into groups who avoided the tax during the first year after the start of business, during the second year, and more than two years later.

All logistic regression models can also be used for prediction.

Logistic regression can only be applied if the part of the one category values of the target variable is not less than 20% and not more than 80% of all observations. However, logistic regression is suitable for relatively general assumptions: independent variables do not necessarily have to be normal; regression errors are not required to be normal; and the homoscedasticity of the target variable is not analyzed.

Li [37] published a logistic regression method for financial reporting fraud detection where key components describing profitability, liquidity, growth prospects, credit status, and shareholder structure are used instead of independent variables. The accuracy of the model reaches 92.86%. The results of research by Devos [38] highlighted many economic, social, psychological, and demographic variables that affect individual taxpayers' behaviour in Australia. Taxpayer representation, the type of tax advice, and the retention or change in tax agents were identified as key indicators. The created model shows that the tax evasion odds ratio is 2.4.

### 4.2.3. Support Vector Machine

The support vector machine (SVM) is a supervised learning technique for classification and regression analysis. This algorithm transforms the initial data into a larger dimension, where a hyperplane that divides the two classes with the greatest possible distance between the classified data is found.

SVMs are practically applied using kernel methods. The ability to learn using the hyperplane is obtained using linear algebra, in which the observations are not directly used; rather, their inner product is. The inner product is calculated by finding the sum of the product of each pair of values of input. For instance, the inner product of input vectors $(a, b)$ and $(c, d)$ would be $(a{\cdot}c) + (b{\cdot}d)$. The prediction of the inputs is made using the dot product of input $(x)$ and support vector $(x_i)$ that is calculated by

$$f(x) = B_0 + sum(a_i{\cdot}(x, x_i)),\qquad(14)$$

where the inner product of input $(x)$ would be calculated with all the support vectors in the data and the coefficients of $B_0$ and $a_i$ (for input) should be estimated using a learning algorithm while training.

The support vector machine, as with other automated classification algorithms, allows:

- Identifying objects (such as taxpayers, invoices, and others) according to their potential dependence on a particular classification group;
- Assigning items (such as taxpayers, and invoices) to a specific category, such as low, medium, and high-risk groups;
- Predicting future events, such as tax evasion, according to the created model.

This method has four main advantages:

- First, it has a control parameter that avoids overfitting;
- Second, it uses the kernel property, so it can be built using problem-related expert knowledge;

- Third, the support vector machine method, as with other effective methods, defines the problem of convex optimization (not a local minimum);
- Finally, SVM approximates the level of test error within the boundary.

The drawback is that the theory only determines the choice of regulation and kernel parameters and the function of the kernel itself. Unfortunately, kernel models can be quite sensitive to retraining.

Data mining techniques ensure the ability to filter the discrepancies in VAT declarations. The scientific paper by Wu et al. [27] presents data mining methods capable of identifying possible VAT discrepancies that may be subject to additional audits. The results show that the proposed method improves the detection of tax evasion and can be used effectively. Another work by DeBarr and Wechsler [39] publishes a study on identifying unfair practices in which various classifications evaluate reputation traits.

### 4.2.4. Neural Networks

The operation of neural networks is based on the interaction of neurons, the smallest interconnected constituents. While their features are limited to signal transmission, large networks of multiple neurons can process huge amounts of information quickly and accurately. The neural network is generally a parallel, distributed, and adaptive system capable of identifying hidden dependencies and improving information processing properties based on training.

Specifically, the transformation process for the inputs and outputs in a feedforward artificial neural network with r inputs, a sole hidden layer composed of q processing elements, and an output unit can be summarized in the following formulation of the network output function for the following model:

$$Y = \hat{f}(x, W) = F\left(\beta_0 + \sum_{j=1}^{q} \beta_j G(x'\gamma_j)\right),\tag{15}$$

where

- $x = (1, x_1, x_2, \ldots, x_r)'$ are the network inputs (independent variables), where 1 corresponds to the bias of a traditional model;
- $\gamma_j = (\gamma_{j0}, \gamma_{j1}, \ldots, \gamma_{ji}, \ldots, \gamma_{jr})' \epsilon \Re^{r+1}$ are the weights of the inputs layer neurons to those of the intermediate or hidden layer;
- $\beta_j$, $j = 0, \ldots, q$, represents the hidden units' connection force to those pertaining to output ($j = 0$ indexes the bias unit), and $q$ is the number of intermediate units, that is, the number of hidden layer nodes;
- $W$ is a vector that includes all the synaptic weights of the network, $\gamma_j$ and $\beta_j$, or connections pattern;
- $Y = \hat{f}(x, W)$ is the network output;
- $F : \Re \to \Re$ is the unit activation function and output while $G : \Re \to \Re$ corresponds to the intermediate neurons activation function.

Just as with all automated classification algorithms, neural networks allow identifying objects (such as taxpayers, invoices) according to their potential dependence on a particular class and assigning objects (such as taxpayers, invoices) to a particular group category, such as low, medium, and high risk. Finally, neural networks allow for predicting the model of future events, such as tax evasion.

The learning option makes it possible to significantly shorten the computational time and use smaller resources, reducing the cost of such calculations and increasing work efficiency. Another neural network advantage is the ability to achieve the desired high accuracy (which is due to the result being "grown" and approaching the desired value for the desired length of time), which is often not allowed by other methods. High accuracy of the result achieved due to the long "growing" period leads to a cumbersome result, leading to ineffective use. Another drawback of neural networks is that very accurate

results require large computational resources. As this method is based on a random search, improper selection of initial constants and ranges can result in long calculations and clumsy results. On the other hand, proper selection of initial constants and intervals would not only achieve the same accuracy but would achieve the same accuracy and be faster and simpler.

As mentioned before, the characterization of known taxpayers and the attribution of their characteristic behavior groups make it possible to assess the behavior of others. The denoted authors, Gonzales and Velasquez [24], published a neural networks-based data mining methodology used by the Chilean tax administrator, linking fraud and non-fraud cases with the taxpayer's historical parameters and characteristics. In a similar study by Serrano et al. [40], a constructed neural network is trained with a small data set. Its accuracy, in terms of the coefficient of determination, reaches 0.86.

### 4.3. Methods Based on Dependencies and Logical Relations

The analysis of variability of quantitative 'variables' values often requires determining whether the observed variables are dependent or independent, identifying the tendency of the relationship between them as well as the form of their relation. It can be linear or nonlinear (square, logarithmic). Searching for logical relations in data is one of the most common data mining tasks. Logic rules can explain very complex relations. They allow for predicting and linking different life aspects to the whole. Logic rules often explain political or business forecasts. It is possible to forecast events in separate segments of economic activity. Predicting social behaviour by linking actions with motives, attitudes, demographic characteristics of social groups, and living environment is also possible.

#### 4.3.1. Multidimensional Regression

One of the most commonly used prediction tasks is the evaluation of multidimensional regression (MLR). Regression analysis is divided into parametric and nonparametric, depending on whether the observed random size distribution family is known. The parametric assessment presumes that the regression function describing the data belongs to a certain narrow family of functions that depend on a small number of parameters. Nonparametric assessment requires no parametric assumptions about the function. Instead, other presumptions are made, e.g., concerning the continuity of the function (e.g., that the function has a second-order continuous derivative) or that it is integrated. In addition to the linear least squares or the highest probability methods, nonparametric methods, such as the generalized additive model method, local regression method, smoothing spines, nuclear, and others, are also popular in regression practice.

For the MLR model, the response (dependent) variable y is assumed to be a function of k independent variables $x_i$. The general form of the equation is computed

$$y_i = \beta_0 + \beta_1 x_1 + \beta_2 x_2 + \cdots \beta_k x_k + \epsilon, \tag{16}$$

where $\beta_0$ and $\beta_i$ stand for the fitting constants; $x_i$ represents the $i$th observation of each of the explanatory variables, $y_i$ stands for the ith prediction, and $\epsilon_i$ is a random error term representing the remaining effects of variables on $y$, which are not covered by the model (residuals). The least squares criterion for the minimum sum of squares of error terms is usually applied to determine the fitting constants.

Regression models help in:

- Defining interrelations between different indicators, for example, by examining taxpayers' behavioural habits;
- Predicting future events, such as tax evasion.

The main advantage of linear regression analysis is selecting a function (model) for linking variables. The disadvantage of the least squares and the highest probability regression analysis is that the research results' acceptance and validation require the regression's assumptions to be met. While nonparametric regression model estimation does not satisfy the Gaussian assumption, the model's prediction accuracy is reduced.

The combination of economic, cultural, sociological, psychological, and technical factors influences 'taxpayers' decisions on whether to fulfil their tax obligations. The study published by Poco et al. [41] presents the results of the research based on a regression model, stating that tax evasion is determined by the unevenness of the tax system, the size of the tax burden, the possibility of abusing taxpayers' money and, finally, corruption in the political class. Other authors, Alstadsæter and Jacob [42], published an analysis of the role of tax incentives and tax awareness for tax evasion in Sweden.

4.3.2. Association Rules

The basis for setting association rules is to determine the patterns denoting interrelationships between data. These patterns reveal the internal data structure and regularities inherent to data subsets, expressed in a user-understandable form, i.e., interconnection rules. An important feature of data mining is the non-triviality of the searchable patterns. Patterns must represent unobvious, unexpected data relations, known as hidden knowledge. The rules of associations allow for determining the relations between the regularity of events or processes or, in other words, relating various facts about events.

In the association rules model (ARM), the rules are usually represented in implication rules, such as $x_i \rightarrow x_j$, where both $x_i$, $x_j \subseteq I$ and $x_i \cap x_j = \varnothing$. The left-hand side (LHS) is the antecedent, and the right-hand side (RHS) is the consequent. The strength of association rules is calculated based on two important measures: support and confidence computed

$$s(x_i \rightarrow x_j) = \frac{freq(x_i \cup x_j)}{n}, \tag{17}$$

$$c(x_i \rightarrow x_j) = \frac{freq(x_i \cup x_j)}{freq(x_i)}, \tag{18}$$

where $freq(x_i \cup x_j)$ is the count or frequency of the combination $x_i$ and $x_j$ while $freq(x_i)$ is the count or frequency of $x_i$. Support measures how often the rule applies to the whole sample. Confidence measures how much confidence that the rule holds.

Searching for association rules, such as other dependency testing methods, offers an alternative way to quickly find the required information when processing large amounts of data and detecting relations between them. Association rules are especially relevant for gaining knowledge about taxpayers' behaviour and its relationship with tax administration control and monitoring activities.

When determining association rules, all the analyzed elements are considered the same (homogeneous), i.e., have the same attributes. Determining the number of association rules is a key issue. Often, many association rules can be found that may be too obvious or useless, i.e., not interesting. This exploration process is useful for identifying knowledge that is new, previously unknown, non-trivial, practically useful, interpreted, and necessary for decision-making processes.

The characterization of known taxpayers and the attribution of their characteristic groups of behaviour make it possible to assess the nature of the behaviour of others. Gonzales and Velasquez [24] linked fraud and non-fraud cases to taxpayers' historical parameters and characteristics. Wu et al. [27] disclose data retrieval methods that help identify possible non-conformities in the VAT that may be subject to additional audits. The results show that the proposed method improves the detection of tax evasion and can be used effectively.

*4.4. Other Data Mining Methods*

In statistical research, monitoring certain tasks, interpreting situations, making decisions, and others are often necessary. Therefore, methods for data systematization and graphical representation are used. A detailed description of the collected information and data graphs often allow drawing reasonable conclusions about the characteristics of the whole dataset.

4.4.1. The Selection of Important Features

The main idea of feature analysis can be defined as the division of the observed variables into certain groups, considering their intercorrelations and believing that the variables of each group are united concerning some not directly observed features, also called latent features. Before starting feature analysis, the number of possible attributes and the variables that will form groups explaining them is unknown. Therefore, the variables are initially grouped according to their intercorrelations, and then common group features are identified. Thus, the analysis helps explain the correlation between many variables by the impact of common features. In turn, moving from variables to features makes the information more covered. The purpose of feature analysis is to change the characteristic variable set of the observed phenomenon into several features assuring minimal loss of information.

The major application area is the distinction and definition of relations and patterns between variables, such as identifying key features of taxpayers' behaviour or creating a topology for the effectiveness of interrelated actions by the tax administrator.

The following issues are encountered in feature analysis:

- Latent features do not always exist and can not always be reliably distinguished;
- Different sets of feature analysis applied to the same data produce different sets of possible features;
- Excluded features are not always easy to interpret.

Typically, feature analysis can help decide how many latent attributes would be sufficient to explain the dependency structure of investigated variables, defining the types of features and how well they explain the analyzed data.

The quality of tax-related restructuring depends on the efficiency of the tax administrator in ensuring the practical collection of government revenue. In their publication, Siti and Bojuwon [43] examine the psychometric properties of tax services for the efficiency of tax administration for self-employed taxpayers. In other words, significant features that help the tax administrator ensure tax system effectiveness is distinguished. Research-based feature analysis conducted by Moeinadin et al. [44] concluded that functional dependencies associated 62.7% of tax obligations with social, individual, structural, and legal features.

4.4.2. Anomaly Detection

The methods for determining whether the created model properly describes the available data are called the diagnostics of the model that is being developed. Diagnostic methods can reveal emerging problems and suggest ways to overcome them. Anomaly detection combines ways to identify "unusual", "strange," or perhaps just false records (observations). When analyzing one or two variables, it is not difficult to identify such data—it is enough to draw their graphs (e.g., a spreading chart)—but when the number of variables is "high", it would be preferable to have automated diagnostic procedures.

"Unusual" model data are divided into outliers, high-leverage, and influential data:

- Model outliers are the dependent variable values that are "sharply" different from the values predicted by the model. This deviation is usually measured by standardized model errors, which can be defined differently. One way would be to rely on the values of the diagonal elements of the hat matrix;
- Observations that are "far away" from the "centre" of predictive variables (i.e., have a high leverage) may have a potentially greater influence on regression coefficients. One of the most popular characteristics of observation influence is its hat value;
- High-leverage observations, which also have the feature of exclusivity, are influential because, by removing them from the model, the coefficients significantly change. Cook's distance is often used to measure the change.

In practice, anomalies can be detected in various tasks associated with recording unusual activities, such as detecting fraud cases or performance monitoring.

Statistical anomaly detection methods are mathematically based, and if a probabilistic model is known, the methods are highly effective and help reveal the meaning of the depicted anomalies. In addition, the built-in models are often quite compact, allowing for the detection of abnormal data without storing original datasets, which usually require a lot of space. However, the advantages of these methods limit their practical application: prior knowledge of the distribution of the data set is required. One distribution cannot describe all the data because the data may come from multiple distributions. Thus, with large-scale data, the identification of anomalous members becomes a solution to a large number of tasks.

Anomalies show a very extreme result compared to most data set cases. Income tax declarations were analyzed by applying stratified samples in the study by Levin [23] of US individual taxpayers. The number of members in different layers ranged from 0.84% to 1.19%. However, in some cases, the impact of these members was particularly high when assessing the numerical characteristics of the whole sample. Gonzales and Velasquez [24] showed that primary data processing is inextricably linked to detecting anomalous members.

### 4.5. The Application of Clustering and Classification Methods in the Field of Taxpayer Control and Monitoring

The above-described analytical methods for taxpayers' segmentation and risk assessment can be used to solve various tax administration tasks: they allow performing 'taxpayers' segmentation, analyzing taxpayers' behaviour, assessing taxpayers' riskiness as well as the tendency to non-registration, undeclaration, and tax evasion risks.

By applying the denoted methods and software, analytics can be adapted at the level of aggregated analysis of the conceptual risk management model (Figure 3). At that level, the information available on taxpayers is analyzed to arrange a large set of well-known or unrelated data and draw conclusions about taxpayers' behaviour, similarities, and differences. These results would be used for subsequent actions, such as monitoring one or another segment and selecting and applying control actions. These actions can already be directed to a case study, i.e., into the case analysis square (Figure 3). For example, based on the results of the scientific analysis of the tendency to avoid tax of the taxpayers 'groups' (i.e., unfair use of 0 per cent VAT relief), the tax administrator may decide to initiate tax audits in a particular group of taxpayers identified during the analysis.

It is important to note that conducting data analysis with the help of smart tools enables the tax administrator to perform taxpayer monitoring and control actions "today" and take preventive actions. Without such tools and techniques, most control actions start too late, i.e., after the event, such as clearing specific tax evasion cases in a particular economic sector.

### 4.6. Model Selection Criteria

To evaluate the model's prediction, *AIC*, *ASE*, *SBC*, *MSE*, and *RMSE* are selected as the evaluation criteria of the model in this paper. They are calculated according to formulas

$$AIC = 2k - 2\ln(L), \tag{19}$$

$$ASE = \frac{\sum_{i=1}^{n}(\hat{y}_i - y_i)^2}{n}, \tag{20}$$

$$SBC = k \cdot \ln(n) - 2\ln(L), \tag{21}$$

$$MSE = \frac{1}{n}\sum_{i=1}^{n}(\hat{y}_i - y_i)^2, \tag{22}$$

$$RMSE = \sqrt{MSE}, \tag{23}$$

where $k$ is the number of independent variables, $L$ is the log-likelihood estimate, $n$ is sample size, $y_i$ represents the true value, and $\hat{y}_i$ is the predicted value.

## 5. Experimental Results

Segmentation and risk assessment models, created during the implementation phase, were based on hypotheses raised with the tax administrator. They can be categorized into separate groups: micro-analysis (taxpayer segmentation, VAT gap, and fake declarations), macro-analysis (tax revenue forecasting, tax calendar optimization), and behavior (tools oriented to increase the share of legal consumption of excise goods, measures to encourage individuals to demand purchase documents from legal persons). All the solved problems are implemented cyclically in Figure 8.

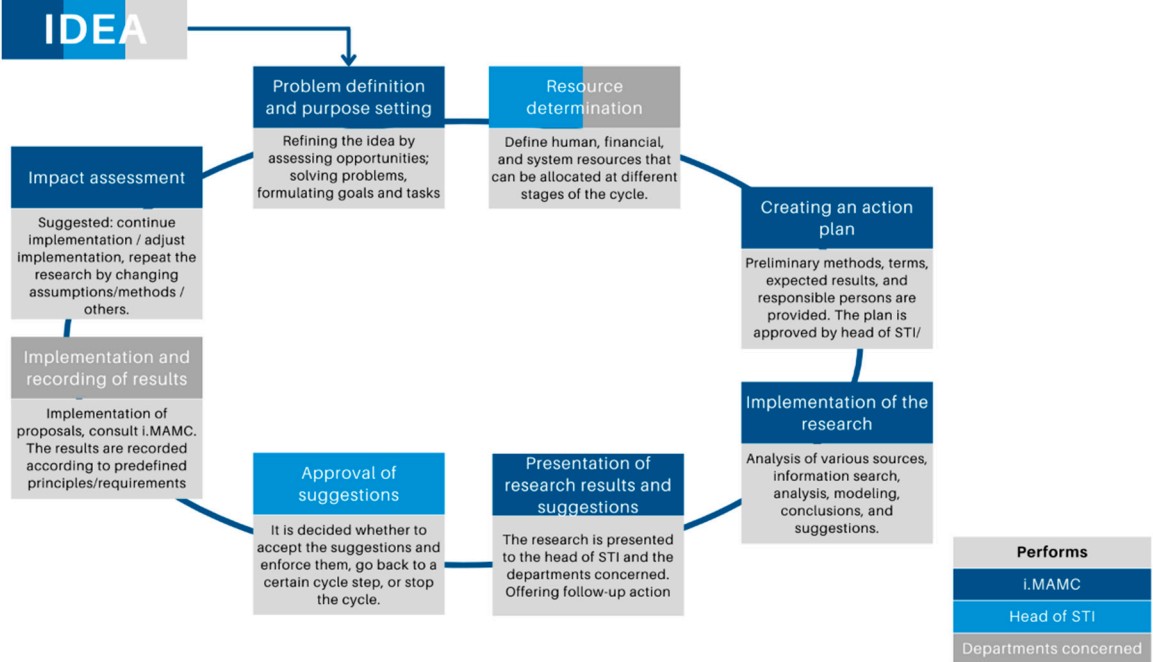

**Figure 8.** Diagram of cyclical implementation of tasks.

Two examples of the developed models are presented below. Due to confidentiality restrictions, the details of the data used and a detailed description of the results are not included.

### 5.1. Segmentation Model Based on Taxpayers' Behaviour

This model aims to discover and understand existing taxpayer segments, where taxpayers are as homogeneous as possible within the segment while as heterogeneous as possible between the segments. The model helps the tax administrator to better disclose the characteristic behaviour of taxpayers' segments (their groups) and enables further detailed analysis of these segments, resulting in the identification of factors affecting the behaviour of a particular segment.

Since the entire set of taxpayers is non-homogeneous and taxpayer behavior is highly dependent on many attributes, applying the model to the subset of taxpayers with similar cornerstone features is only appropriate. This avoids the natural division of taxpayers into segments based on one essential difference or similarity, such as the assignment to individuals or legal persons, and the pursuit or non-execution of economic activity.

A subset of taxpayers, subject to a segmentation-based behaviour model, must be clearly and unambiguously distinguished from the entire taxpayer set so that accidental, inaccurate, or subordinate taxpayers do not distort segmentation results.

The modelling sample includes all Lithuanian legal entities registered as VAT payers. The analysis uses data from the period of one calendar month.

Over 200 hierarchical, geometric, and probabilistic clustering models were developed during the simulation. Finally, models that demonstrated the clearest segmentation results were selected for the tax administrator.

The modelling process was performed according to the recommended SEMMA methodology. SEMMA describes the simulation steps in the following sequence: starting with a statistically representative sample, statistical or visual analysis is applied, then variables are selected and transformed, and, ultimately, models are created and evaluated. The process can be repeated twice as necessary to obtain a satisfactory solution.

### 5.1.1. Variable Selection

The purpose of this step is to select significant variables according to the objectives of the segmentation model. Selected variables must meet the following general requirements:

- From a business perspective, i.e., according to the intended use of the model, the selected variables must be meaningful;
- Variables influence the structure of the segments (verified by preliminary analysis of the created model);
- Time-stable variables.

Various methods and recommendations can be used to select significant variables. Principal component analysis (PCA) method can be used for data quality checks, i.e., for the initial assessment of the significance of optional business indicators. If the objective of segmentation is a behavioural segmentation of general payers, it is recommended to use continuous and discrete behavioural indicators. The variables should be selected according to population and frequency (rejecting the indicators found in <1% of the population). For this reason, it is necessary to examine the characteristics of the broad distribution. Correlating indicators do not necessarily have to be eliminated because several segments have similar behaviour in some situations but differ in their 'weight' correlating indicators.

Kolmogorov's statistics can be used to check whether the variables are distributed in different periods, i.e., to check the variables' stability concerning time:

- If the value of the significance level corresponding to the Kolmogorov statistic is less than 0.05, then the indicator is rejected;
- If the significance level of a Kolmogorov statistic corresponding to the indicator is between 0.05 and 0.1, the indicator can be included or rejected—the decision must be made according to other criteria;
- If the significance level of Kolmogorov's statistic is more than 0.05, the indicator must be added to the modelling.

### 5.1.2. Variable Transformations

Before clustering is performed, it is recommended to use various transformations of variables that perform rotation showing the highest variability in the data. Variable value scales are very important for segmentation. If one of the variables involved in the modelling has values several times larger than all other variables, then the clustering algorithm will immediately increase the impact of that variable. Therefore, unifying the value scales of all variables involved in the simulation is advisable before performing it.

Transformation of distribution value normalizing may be useful for transforming variables whose values' distribution is asymmetric, displaced to either side or has multiple pitches. Standardization-transformed variable mean is equal to 0, and variation is equal to 1. Standardization was used for the segmentation of the taxpayers' population.

After selecting the significant variables and performing the transformations of the variables, eliminating outlier observations is performed before modelling. The quantum and standard deviation methods can be used to accomplish this task. Regarding taxpayer segmentation, about 14.6% of all observations that deviated from the mean value by more than three standard deviations (3-sigma rule) were removed from the sample.

### 5.1.3. Modelling Results

Common characteristics of a successful model:

- About 20 significant variables have to be selected for modelling (in the case of the sample presented in this study, the main components were used);
- Segment structure should consist of 4–7 segments;
- The segments must be homogeneous according to the selected attributes and differ from each other;
- For better understanding and recognition, segments should be 'measurable' in size and identified by their characteristics;
- It must be possible to write the characteristics of each segment in one sentence.

The initial estimation of the generated model is usually conducted by applying a "segmentation map", which is used to calculate the average of the indicators for the entire population, and the average for each segment separately. The significance of the indicators can then be interpreted according to the selected condition: for example, if the average of the indicator for the segment is 30% higher or lower than the average of the indicator for the whole population, a such indicator can be used for the description of the segment, and it is considered interesting from the business side. For segmentation maps, averages are calculated for the metrics on which the segment model is based and for other existing analytical base table indicators are analyzed. The segmentation map examines whether segments are similar or different in their characteristics (significant indicators) or whether the indicators seem meaningful to the segment's interpretation from the business side. The initial estimation of the generated model is also performed using the distances between the clusters.

During the modelling phase, segmentation models were developed by selecting different sets of significant variables and modifying other modelling parameters.

Cluster sizes and interactions between the clusters of the VAT segmentation model are depicted below (Figure 9 and Table 1).

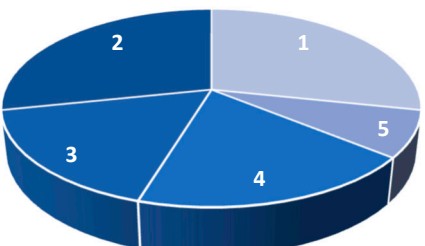

**Figure 9.** Graph of cluster sizes.

**Table 1.** Distance between clusters.

| SEGMENT ▲ | 1 | 2 | 3 | 4 | 5 |
|---|---|---|---|---|---|
| 1 | 0 | 8.010 | 6.687 | 5.200 | 7.345 |
| 2 | 8.010 | 0 | 4.845 | 5.141 | 11.841 |
| 3 | 6.687 | 4.845 | 0 | 4.127 | 10.613 |
| 4 | 5.200 | 5.141 | 4.127 | 0 | 9.880 |
| 5 | 7.345 | 11.841 | 10.613 | 9.880 | 0 |

Due to confidentiality restrictions, the profiles and descriptions of the created clusters are not presented.

### 5.2. VAT Assessment Model for VAT Payer Checkout

This model aims to identify VAT payers who will no longer fulfil their tax obligations within 2 'months of the change in relevant information about them, i.e., they will deregister from the VAT payers' register. Possible causes of substantial information changes are:

1.　Decrease in share capital;
2.　Legal status change (e.g., liquidation, bankruptcy, and reorganization);
3.　Change in the head of the company;
4.　Significant change (decrease) in the number of employees;
5.　Significant decrease in revenue;
6.　Significant decrease in assets;
7.　Etc.

It should be noted that, due to confidentiality restrictions, a list of essential criteria used by the tax administrator in its activities is not included.

Modelling sample—all Lithuanian legal entities registered as VAT payers. At the time of prediction analysis, these individuals must be subject to fulfilling their reporting obligations.

Over 150 neural networks, decision trees, logistic regression, association rules, support vectors, and discriminatory analysis models were developed during the simulation. As a result, the final models showing the best results of the taxpayers registered for VAT were selected for the tax administrator.

The modelling process was performed according to the methodology recommended by SEMMA. The SEMMA describes the simulation steps: starting with a statistically representative sample, applying statistical analysis, selecting variables, and creating and evaluating the models. The process can be repeated multiple times to obtain a satisfactory solution.

### 5.2.1. Analytical Data Flow Process

A set of analytical indicators, based on two months, was used to solve the classification task. Predicted indicator: the attribute showing whether the VAT taxpayer has registered out of VAT payers within two months after a major event, without the initiative of the tax authorities.

**Sampling**: A representative sample selection method, namely separate sampling was used to solve the prediction task. The sample included 808 deregistered and 1616 registered taxpayers.

**Data partitioning**: To create the most appropriate model, the sample is randomly (by applied stratification) partitioned into training and validation subsets. A total of 564 deregistered and 1130 registered taxpayers were enrolled for training, and, respectively, 244 and 486 taxpayers for validation.

**Selection of variables**: according to the model's purpose, this step's objective is to select significant variables. In general, they should meet the following requirements:

- be selected from a business point of view, i.e., significant variables according to the intended use of the model;
- variables should have a significant impact on the prediction result;
- be time-stable variables.

### 5.2.2. Significance and Efficiency of the Model

After applying several different methodologies for predicting VAT taxpayers' behaviour, namely deregistration within two months, the logistic regression model showed the best results. In this model, insignificant variables are omitted because we want to ensure a more accurate estimation of the regression coefficients. To obtain a more accurate model, we used AIC, ASE, SBC, MSE, and RMSE criteria (see Table 2). All parameters of the obtained model are statistically significant ($p$-value < 0.01). The likelihood ratio statistic was used to compare the logistic regression and null models. Due to the low $p$-value (<0.0001), the null hypothesis that the null and full models do not differ was rejected. Therefore, it is concluded that cases of essential information change have a significant impact on VAT payers' who will no longer fulfill their tax obligations within two months.

**Table 2.** Model selection criteria.

| Fit Statistics | Train | Validation |
| --- | --- | --- |
| AIC | 1810.56 | |
| ASE | 0.176804 | 0.190389 |
| SBC | 1870.34 | |
| MSE | 0.177858 | 0.190389 |
| RMSE | 0.421733 | 0.436336 |

Capture response (*CR*) and lift efficiency measures are commonly used to create and select models (see in Figures 10 and 11). The prediction model assigns higher deregistration probabilities to taxpayers who tend to that. Therefore, after sorting the list of predictions in descending order, taxpayers with the highest probability will be at the top. If the model works well, the first decile should include more taxpayers than random sampling in the same volume sample. *CR* is calculated using the formula:

$$CR_{X\%} = \frac{the\ number\ of\ taxpayers\ checked\ out\ between\ the\ top\ X\%}{the\ number\ of\ taxpayers\ registered\ in\ the\ entire\ population} \times 100\%. \quad (24)$$

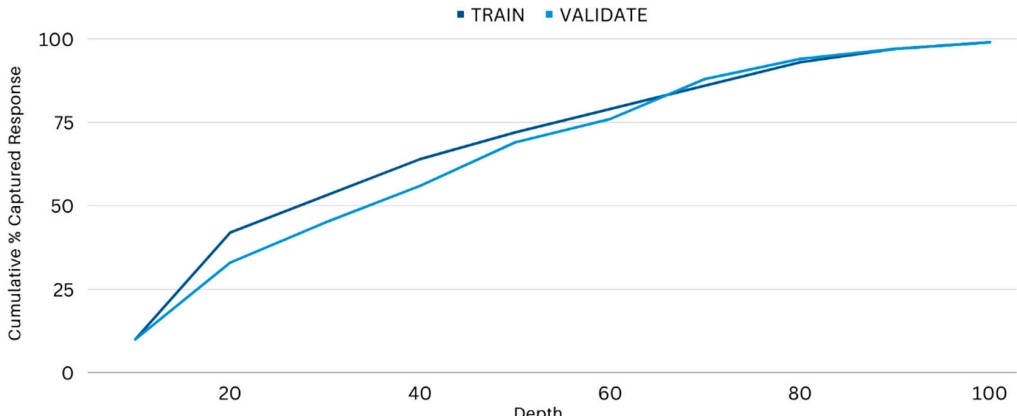

**Figure 10.** CR chart of statistical changes.

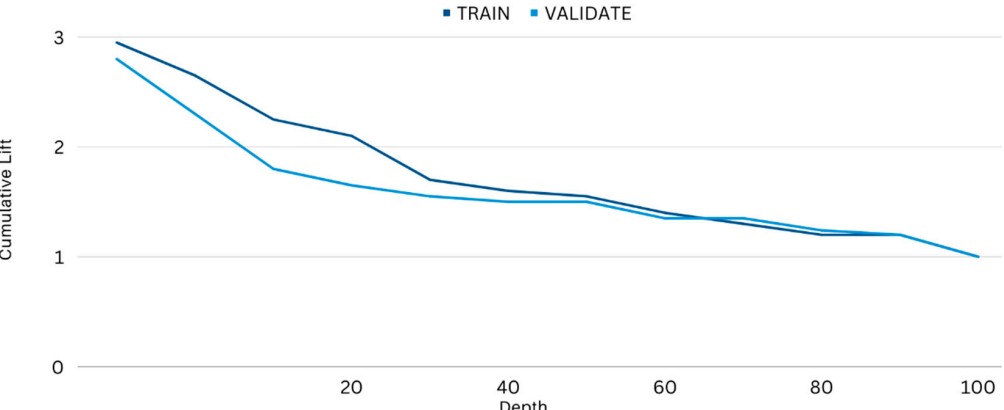

**Figure 11.** LIFT chart of statistical changes.

As it was mentioned before, lift is another model efficiency measure. This measure denotes how much better the model is compared to the random selection of taxpayers. In a random sample containing 10% of all taxpayers, there should be 10% of all deregistered taxpayers. So if the first decile contains 20% of the total number of deregistered taxpayers, then the lift value is equal to 2 because, in a random sample with 10% of all taxpayers, there

should also be about 10% of all deregistered taxpayers. The lift value can be calculated using the following formula:

$$lift_{X\%} = \frac{the\ number\ of\ taxpayers\ checked\ out\ between\ the\ top\ X\%}{expected\ number\ of\ taxpayers\ checked\ outis\ X\%\ of\ the\ population} \times 100\%. \quad (25)$$

In the first decile (see in Figure 10), the CR for the training sample reaches 26% and exceeds 22.5% for the test sample. Respectively (see in Figure 11), the lift statistics value for the training sample shows that, compared to the random sample of taxpayers, the model is better than 2.6 times, while for the validation subset, the same model for taxpayers' deregistration from the VAT payer register is better than 2.25 compared to random sampling.

## 6. Conclusions and Future Works

The clustering and classification methods used to describe taxpayers' historical fiscal behaviour in fraud and non-fraud suggest that specific individual characteristics can be identified to distinguish between groups.

The study's findings reveal that the logistic regression method used in cases where the result of the VAT payer's deregistration without the initiative of tax authorities was known as a reliable way of identifying significant variables that can predict the future evasion of tax obligations. Analyzing the distribution of variable values, it is identified that this situation is due to the behavioural changes that affect the appearance of more extreme values.

Present findings confirm that logistic regression and multilayer neural network techniques show the best results as detection models, the latter being slightly less accurate. Concerning training and validation samples, the percentage of correctly identified VAT deregistration cases within two months was 82.3% and 81%, respectively. In the first decile, it was, respectively, better 2.6 and 2.25 times, compared to random sampling.

Considering the obtained results and the fact that in practice, only a small number of company tax audits can be carried out, it is recommended to combine the results of analytical methods with the audit results, thus providing feedback on the training of the models.

Finally, according to the publicly disclosed data of the Lithuanian tax administrator, after introducing taxpayers' risk assessment models, the rates of the shadow economy began to drop. According to the report of the European Commission, the Center for Social and Economic Research on the VAT gap in the countries of the European Union (TAXUD, 2022), in 2020, the VAT gap in Lithuania amounted to 19.3% and compared to 2019 it decreased by 1.6%. In 2021, it amounted to 14.3%, i.e., 5% less than in 2020. As announced, an action plan for reducing the shadow economy and the VAT gap is being implemented by making tax administration measures more effective and fighting against the shadow economy. A constituent part of this plan is the taxpayer risk assessment models presented in this article and applied by the tax administrator. Data analysis and insights related to the expansion of the shadow economy phenomena in Lithuania and proper identification of the causes and ways to minimize it allow the experts to predict that the shadow economy decrease will continue.

Future research should consider the potential ability to complement the historical behavioural information used in data discovery methods with the analysis of social networks. This assumption might improve the meaningful identification of taxpayers' risks predicting tax avoidance through social network analytics and should be addressed in future studies.

**Author Contributions:** Conceptualization: T.R., L.K., E.S. and J.A.; methodology: T.R., L.K., E.S. and J.A.; validation: T.R.; resources: T.R.; data curation: T.R.; writing— L.K., E.S. and J.A.; writing—review and editing T.R., L.K., M.L., E.S., M.F. and J.A.; supervision: T.R. All authors have read and agreed to the published version of the manuscript.

**Funding:** This research received no external funding.

**Institutional Review Board Statement:** Not applicable.

**Informed Consent Statement:** Not applicable.

**Data Availability Statement:** Taxpayer data is confidential information.

**Conflicts of Interest:** The authors declare no conflict of interest.

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
