# Peer review of "Tax Fraud Reduction Using Analytics in an East European Country"

_axioms, doi:10.3390/axioms12030288_

Round 1

Author Response

Comments and Suggestions for Authors

The present manuscript deals with the problem tax evasion of many companies in European Union countries. The main object of the work is to increase the efficiency of the detection of tax evasion by applying data mining methods. By virtue of standard data mining techniques are developed to address segmentation, risk assessment, behavioral templates and tax crime detection. They provide a realistic detection of tax evasion and are able to extract hidden knowledge from original dataset.

The literature review and the presentation of the adopted methodologies are clear and detailed, and the paper is well written overall. However, the work presents some (major) issues needing attention.

  • The authors have to specify better the innovations of their contribute. In particular, they have to insert also some mathematical aspects of the presented algorithms. This Referee does not see no formula throughout the paper (except those of Fig. 10, 11, which are not clear). The absence of any mathematical formalism does not fit the standard of such journal.

 Response: The mentioned formulas were rewritten. The description of the methods is supplemented with formulas.

  • In order to enforce the statistical analysis, the author must introduce some well-known metrics well as the RMSE or the MAPE to assess the goodness of their validation.

 Response: Section 5.2.2 is supplemented with model significance using AIC, ASE, SBC, MSE and RMSE criteria.

Reviewer 2 Report

Overall, I like the paper because the research has something potential to contribute to the literature. At the same time, it is my opinion that the paper has not been sufficiently vetted. Below I provide two examples:

First, section heading 4.2.1 Decision trees is the same as the section heading 4.2.2. Similarly, the section heading 4.2.4 Neural Networks is the same as the section heading 4.3.1 or 4.3.2. Under these section headings, the authors discuss substantially different issues. 

Second, in section 6: Conclusions and future works section authors write, "The findings of the study reveal that logistic regression method used in cases where the results of the VAT payer's deregistration without the ..."  But I could not find the results from logistic regression models presented in the main body of the paper.

On my part as a critical reviewer, these two issues raised an uneaseness with the overall quality of the paper.

Author Response

Comments and Suggestions for Authors

Overall, I like the paper because the research has something potential to contribute to the literature. At the same time, it is my opinion that the paper has not been sufficiently vetted. Below I provide two examples:

First, section heading 4.2.1 Decision trees is the same as the section heading 4.2.2. Similarly, the section heading 4.2.4 Neural Networks is the same as the section heading 4.3.1 or 4.3.2. Under these section headings, the authors discuss substantially different issues. 

Response: Technical errors was corrected. The sections headings was clarified:

  • The title of section 4.2.2 was changed to "Logistic regression".
  • The title of section 4.3.1 was changed to "Multidimensional regression".
  • The title of section 4.3.2 was changed to "Association rules".

Second, in section 6: Conclusions and future works section authors write, "The findings of the study reveal that logistic regression method used in cases where the results of the VAT payer's deregistration without the ..."  But I could not find the results from logistic regression models presented in the main body of the paper.

Response: We supplemented the beginning of section 5.2.2 with results that show the significance of the logistic regression model.

On my part as a critical reviewer, these two issues raised an uneaseness with the overall quality of the paper.

Response: We have improved the manuscript based on your comments.

Round 2

Reviewer 1 Report

The authors have improved the paper second this referee's comments, so that it can be accepted in its current form.

Reviewer 2 Report

It is my policy not to review the same paper twice so that the authors need not face "double jeopardy" (It will be unfair for the authors that their research paper being rejected from the same reviewer).